# Just Seeing Is Not Enough for Believing: Immunolabelling as Indisputable Proof of SARS-CoV-2 Virions in Infected Tissue

**DOI:** 10.3390/v13091816

**Published:** 2021-09-13

**Authors:** Andreja Erman, Karmen Wechtersbach, Daniel Velkavrh, Jerica Pleško, Maja Frelih, Nika Kojc

**Affiliations:** 1Institute of Cell Biology, Faculty of Medicine, University of Ljubljana, 1000 Ljubljana, Slovenia; andreja.erman@mf.uni-lj.si; 2Institute of Pathology, Faculty of Medicine, University of Ljubljana, 1000 Ljubljana, Slovenia; karmen.wechtersbach@mf.uni-lj.si (K.W.); daniel.velkavrh@mf.uni-lj.si (D.V.); jerica.plesko@mf.uni-lj.si (J.P.); maja.frelih@mf.uni-lj.si (M.F.)

**Keywords:** SARS-CoV-2, immunolabelling, light microscopy, electron microscopy, correlative microscopy

## Abstract

Background: There is increasing evidence that identification of SARS-CoV-2 virions by transmission electron microscopy could be misleading due to the similar morphology of virions and ubiquitous cell structures. This study thus aimed to establish methods for indisputable proof of the presence of SARS-CoV-2 virions in the observed tissue. Methods: We developed a variant of the correlative microscopy approach for SARS-CoV-2 protein identification using immunohistochemical labelling of SARS-CoV-2 proteins on light and electron microscopy levels. We also performed immunogold labelling of SARS-CoV-2 virions. Results: Immunohistochemistry (IHC) of SARS-CoV-2 nucleocapsid proteins and subsequent correlative microscopy undoubtedly proved the presence of SARS-CoV-2 virions in the analysed human nasopharyngeal tissue. The presence of SARS-CoV-2 virions was also confirmed by immunogold labelling for the first time. Conclusions: Immunoelectron microscopy is the most reliable method for distinguishing intracellular viral particles from normal cell structures of similar morphology and size as virions. Furthermore, we developed a variant of correlative microscopy that allows pathologists to check the results of IHC performed first on routinely used paraffin-embedded samples, followed by semithin, and finally by ultrathin sections. Both methodological approaches indisputably proved the presence of SARS-CoV-2 virions in cells.

## 1. Introduction

Since SARS-CoV-2 emerged, many studies have reported alleged SARS-CoV-2 virions in COVID-19 patients’ tissues found by transmission electron microscopy (TEM) [1,2,3]. Detection of viral particles by TEM only was used as sufficient evidence for viral identification in the aforementioned studies, indicating evidence of direct viral infection of certain cells and tissues. However, few authors have drawn attention to the possible misinterpretation of these findings due to the similarity of structure and size between virions and normal cell structures [4,5]. In a previous study, we showed cell structures resembling SARS-CoV-2 virions in lung and kidney specimens of SARS-CoV-2 positive and negative patients based on their RT-PCR results for SARS-CoV-2 RNA [6]. In light of our results, we therefore suggested that for unambiguous identification of virions in cells and tissues, methods such as immunohistochemistry or immunoelectron microscopy are needed, in addition to viral genetic material analysis.

Existing methods used for viral detection in tissue specimens have known limitations. For example, immunohistochemistry can be falsely positive or negative due to several reasons in the pre-analytical or analytical phase. Furthermore, conventional TEM analysis strongly depends on tissue preservation, which might distort the structure of viruses if it is not performed optimally. An additional disadvantage of TEM analysis is that the sample might miss the area containing viruses [7]. On the other hand, immunoelectron microscopy is a very powerful and highly sophisticated method for the detection and identification of viruses, but is mostly available in academic settings. The method that connects the aforementioned methods and also represents a bridge between light and electron microscopy is correlative microscopy. 

We performed a variant of correlative immunohistochemistry and immunogold labelling of SARS-CoV-2 proteins for indisputable proof of the presence of SARS-CoV-2 virions in nasopharyngeal tissue of patients with positive RT-PCR for SARS-CoV-2 RNA. We combined immunohistochemical labelling of SARS-CoV-2 proteins on light and electron microscopy levels and thus developed a correlative microscopy approach for SARS-CoV-2 protein identification, for which there are no reports in the literature to date. We also showed that a specific and reliable immunoreaction crucially depends on the immunolabelling protocol, from the first steps of tissue sample preparation to the choice of antibody. We believe that immunolabelling, if performed correctly, is the most reliable method of distinguishing intracellular viral particles from normal cell structures of similar structure and size to virions. To the best of our knowledge, this is the first report of the immunolabelling of SARS-CoV-2 virions on an electron microscopy level.

## 2. Materials and Methods

### 2.1. Patients

Autopsy specimens from nasopharyngeal tissue of SARS-CoV-2 positive patients by RT-PCR were taken for immunohistochemistry and immunogold labelling of SARS-CoV-2 virions.

### 2.2. Immunohistochemistry (IHC) for Correlative and Transmission Electron Microscopy (TEM)

Nasopharyngeal tissue pieces (5 mm^3^) were fixed in 4% paraformaldehyde (Merck, Darmstadt, Germany) for 24 h, dehydrated in ethanol, and embedded in paraffin. The tissue sample for immunohistochemistry was obtained by punching the paraffin block with a manual tissue punch. Punched paraffin blocks were then deparaffinized and rehydrated with ethanol. Antigen retrieval was performed with microwave heating in 10 mM sodium citrate buffer, pH 6.0 (Sigma Aldrich, Burlington, MA, USA) for 20 min. Endogenous peroxidases were blocked by 3% hydrogen peroxide (Ventana, Monterey, CA, USA) for 15 min following incubation in primary rabbit antibodies against SARS-CoV-2 nucleocapsid protein (1:600; NB100, Novus Biologicals, UK) for 2 h at room temperature. After washing with reaction buffer (Ventana, Monterey, CA, USA), an UltraView Universal DAB Detection Kit (Ventana, Monterey, CA, USA) was used with the following protocol: HRP enzyme conjugated to goat anti-rabbit secondary antibodies for 20 min, reaction buffer (Ventana, Monterey, CA, USA) twice for 5 min, a mixture of 3,3’-diaminobenzidine (DAB) chromogen and H_2_O_2_ for 20 min, and finally washed with reaction buffer (Ventana, Monterey, CA, USA) and distilled water for 5 min. 

After IHC, tissue samples were transferred to 0.1 M Millonig’s phosphate buffer overnight and processed the next day for transmission electron microscopy. They were first post-fixed in 1% OsO_4_ (Merck, Darmstadt, Germany) for 30 min, then dehydrated in graded concentrations of ethanol and propylene oxide 1-2-propylene oxide (Merck, Darmstadt, Germany) for 10 min in each solution. Tissue pieces were then incubated in a mixture (1:1) of 1-2-propylene oxide and Epon 812 resin (Serva Electrophoresis, Heidelberg, Germany) for 20 min, embedded in 100% Epon, and polymerized at 60 °C for 24 h. The next day, semithin sections of 1 µm were cut with an ultramicrotome Leica EM UC6 and stained with Azure II staining solution to find SARS-CoV-2-positive cells as the region of interest, to select this region for ultrathin sectioning. Ultrathin sections of 60 nm were cut and viewed in a transmission electron microscope JEM- 1200 EXII (JEOL, Tokyo, Japan) at 60 kV (Figure 1).

### 2.3. Immunogold Labelling on Ultrathin Sections

For immunogold labelling on ultrathin Epon sections, small samples (2 mm^3^) of nasopharyngeal tissue were fixed in a mixture of 4% paraformaldehyde (Merck, Darmstadt, Germany) and 2% glutaraldehyde (Serva, Heidelberg, Germany) in 0.2 M cacodylate buffer (pH 7.3) for 3 h at 4 °C. Overnight rinsing in 0.33 M sucrose in 0.2 M cacodylate buffer was followed by post-fixation with 1% OsO_4_ (Serva, Heidelberg, Germany) for 1 h. After dehydration in an ethanol series, tissue samples were embedded in Epon 812 resin (Serva Electrophoresis, Heidelberg, Germany) and cut into ultrathin sections (60 nm thick), which were collected on nickel grids. Antigen unmasking treatment was performed by incubation of some grids in a humid chamber on a large drop of a saturated aqueous solution of sodium meta periodate (Merck, Darmstadt, Germany) for 1 h at room temperature. Grids were then washed in distilled water. Non-specific labelling was blocked by blocking buffer (5% foetal calf serum in 0.1% BSA in PBS (washing buffer)) for 30 min at room temperature. Primary antibodies against SARS-CoV-2 nucleocapsid protein (NB100; Novus Biologicals, UK) diluted 1:50 or SARS-CoV-2 spike glycoprotein S1 (ab 275759; Abcam, Cambridge, UK) diluted 1:100 was applied and incubated overnight at 4 °C. After washing in washing buffer, goat anti-rabbit secondary antibodies with 18 nm gold (Au) diluted 1:40 in blocking buffer were applied for 90 min. In all cases, the manufacturers of the antibodies used provided proof of validation of the technical specifications. In all cases, negative controls were also performed, in which primary antibodies were replaced with PBS. Sections were counterstained with uranyl acetate and lead citrate. Ultrathin sections were viewed on a Philips CM100 transmission electron microscope (Philips, Amsterdam, The Netherlands) at 80 kV (Figure 1).

For immunogold labelling on ultrathin Lowicryl sections, small cubes (1 mm^3^) of nasopharyngeal tissue were transferred to 2% paraformaldehyde (Merck, Darmstadt, Germany) plus 0.05% glutaraldehyde (Serva, Heidelberg, Germany) in PBS for 1 h at room temperature. Samples were dehydrated by progressively lowering the temperature, then embedded in Lowicryl HM20 resin (Polysciences, Eppelheim, Germany) in a Leica AFS apparatus (Leica Microsystems, Vienna, Austria), according to the following protocol: 30% ethanol for 30 min at 0 °C, 55% ethanol for 30 min at −15 °C, 70% ethanol for 30 min at −30 °C, 100% ethanol for 1 h at −50 °C, 75% ethanol/25% HM20 for 1 h at −50 °C, 50% ethanol/50% HM20 for 1 h at −50 °C, 25% ethanol/75% HM20 for 1 h at −50 °C, 100% HM20 for 1 h, and 100% HM20 overnight at −50 °C. HM20 was polymerized for 48 h at −50 °C and then for 24 h at −20 °C under UV light. Ultrathin sections (60 nm thick) were cut and collected on nickel grids. Sections were washed in washing buffer (0.1% Na-azide, 0.8% BSA, and 0.1% IGSS gelatine in PBS), blocked in blocking buffer (5% foetal calf serum in 0.1% BSA in PBS (washing buffer)) for 30 min at room temperature, and incubated overnight at 4 °C with primary antibodies against SARS-CoV-2 nucleocapsid protein (NB100-56576; Novus Biologicals, UK) diluted 1:50 or SARS-CoV-2 spike glycoprotein S1 (ab 275759; Abcam, Cambridge, UK) diluted 1:100. After washing in washing buffer, goat anti-rabbit secondary antibodies with 18 nm gold (Au) diluted 1:40 in blocking buffer were applied for 90 min. Sections were counterstained with uranyl acetate and lead citrate. Ultrathin sections were observed in a Philips CM100 transmission electron microscope (Philips, Amsterdam, The Netherlands) at 80 kV (Figure 1).

## 3. Results

### 3.1. Correlative Microscopy Approach and SARS-CoV-2 Virion Detection

The results of immunohistochemistry of SARS-CoV-2 nucleocapsid proteins and subsequent correlative microscopy undoubtedly proved the presence of SARS-CoV-2 virions in the analysed human nasopharyngeal tissue (Figure 2). We detected cells with positive immunohistochemical staining in semithin sections by light microscopy (Figure 2a). The same cells with positive immunohistochemical staining were examined on ultrathin Epon sections, showing electron-dense material in the cytoplasm inside endosomes (Figure 2b,c). Finally, at higher magnification, we identified in endosomes of the observed cell virions marked with a fine granular electron-dense reaction corresponding to the product of DAB (Figure 2d).

### 3.2. Immunoelectron Microscopy Approach and SARS-CoV-2 Virion Detection

Through immunogold labelling, we confirmed the presence of SARS-CoV-2 virions in human nasopharyngeal tissue samples. The results of this method greatly depended on the type of ultrathin sections and primary antibodies used in the immunolabelling protocol (Figure 3). Immunogold reaction on ultrathin Epon sections with primary antibodies against SARS-CoV-2 nucleocapsid protein (Novus Biologicals) was ubiquitous and therefore unspecific (Figure 3a), while on ultrathin Epon sections with primary antibodies against SARS-CoV-2 spike glycoprotein S1 (Abcam), it was scarce and unspecific (Figure 3b). After an antigen unmasking procedure on ultrathin Epon sections (described in Methods) with primary antibodies against SARS-CoV-2 spike glycoprotein S1 (Abcam), immunogold labelling was more specific but weak (Figure 3c). Immunogold labelling on ultrathin Lowicryl sections with primary antibodies against SARS-CoV-2 spike glycoprotein S1 (Abcam) was specific and intense (Figure 3d).

## 4. Discussion

Viral genetic material analysis of patients’ nasopharyngeal tissue by itself is not sufficient proof of SARS-CoV-2 active infection in body tissues, but a positive RT-PCR test, which is currently an established test for SARS-CoV-2 infection, is a precondition for further analysis of SARS-CoV-2 virus presence in a particular tissue. Likewise, TEM analysis per se is not enough for unequivocal proof of SARS-CoV-2 virions in the tissue of interest due to the similar structure and size of these virions and ubiquitous cell structures [4,5]. Thus, we emphasised in a previous study that after a positive RT-PCR test for SARS-CoV-2 RNA, IHC or even immunoelectron microscopy is needed for the reliable identification of virions in suspected infected tissues [6].

In this study, we present two approaches to immunolabelling on autopsy nasopharyngeal tissue samples of patients with positive RT-PCR for SARS-CoV-2 RNA. We thus performed correlative microscopy in which we combined the advantages of both IHC and electron microscopy to analyse the same specimen. We first performed an immunohistochemical reaction against SARS-CoV-2 nucleocapsid protein with diaminobenzidine (DAB) chromogen on paraffin sections, which is a standard diagnostic method in our pathology laboratory. DAB is oxidised in the presence of peroxidase and hydrogen peroxide, then bound on secondary antibodies, resulting in the deposition of a brown-coloured precipitate at the site of enzymatic activity, visible under light microscopy and also by electron microscopy, especially after post-fixation with osmium [8]. We also performed the same immunohistochemical protocol on deparaffinized small tissue pieces that were then fixed in osmium, embedded in Epon resin, and cut into semithin sections. Finally, the selected area of cells with positive immunohistochemical staining was cut for TEM. The immunohistochemical reaction enabled visualisation of immunohistochemically labelled SARS-CoV-2 virions in infected cells on a semithin section by light microscopy, followed by TEM detection of SARS-CoV-2 virions in the same cell on ultrathin sections. We managed to preserve immunohistochemical labelling of virions with DAB during the TEM embedding procedure and showed that IHC and TEM can be used as analytical methods for unambiguous identification of SARS-CoV-2 virions. 

Immunohistochemistry is a common supplementary method in pathology for diagnostic purposes, but all steps in this technique must be optimised. Antigens are known to be vulnerable at several stages of the immunohistochemical procedure and also during specimen preparation for TEM [9,10]. By carefully selecting protocols for the IHC and TEM procedure and performing all procedures precisely, we preserved good accessibility of the epitopes and also achieved a highly specific reaction on an electron microscopy level.

From our point of view, immunoelectron microscopy is the most reliable method for distinguishing intracellular viral particles from normal cell structures of similar morphology and size as virions. Unfortunately, special and expensive equipment is needed for this methodology. Therefore, we developed a variant of correlative microscopy that allows every routine pathology laboratory to check its results of IHC performed on routinely used paraffin-embedded samples, as well as with a transmission electron microscope without highly developed instruments and sophisticated techniques. Namely, an IHC reaction at a light microscopy level per se could be misleading due to various factors in the pre-analytical phase and misinterpretation of nonspecific staining. Additionally, low magnification and resolution of IHC do not allow pathologists to see virions or be convinced that an IHC reaction colocalizes with virions. Likewise, conventional TEM analysis per se is also not enough for undoubted proof of present virions. Namely, despite the possibility to see the structures (virions) with TEM due to its high resolution, we can misinterpret them. Therefore, only the combination of the localisation of a positive immunoreaction on the structure of interest (i.e., viruses) and simultaneous identification of this structure on an ultrastructural level gives indisputable proof of the presence of virions. We could therefore recommend pathologists in routine pathology laboratories to confirm their results of IHC against the SARS-CoV-2 virus with correlative microscopy described in our manuscript. In our opinion, this is the main scientific output of our work that can be transferred into everyday practice. We are aware that it represents a small piece of the SARS-CoV-2 jigsaw; however, every piece is important in making the whole picture clearer. We cannot afford false positive or false negative pathology reports these days, especially if we have appropriate knowledge and methodology.

In order to avoid pitfalls in immunogold labelling, appropriate fixation and embedding mediums, which can restrict both antibody access and antibody quality, are crucial [11,12]. In our study, the results of immunogold labelling of SARS-CoV-2 virions strongly depended on the selection of the primary antibody and resin. Only on Lowicryl sections and with an appropriate antibody were we able to get specific staining and a satisfactory ultrastructure. Immunogold labelling of coronaviruses has been reported [13], but we believe this to be the first report to date on the identification of SARS-CoV-2 virions with immunoelectron microscopy.

## 5. Conclusions

To conclude, immunoelectron microscopy, if performed correctly, is the most reliable method for distinguishing intracellular viral particles from normal cell structures of similar morphology and size as virions. Furthermore, we developed a variant of correlative microscopy that allows pathologists to check the results of IHC performed first on routinely used paraffin-embedded samples, followed by semithin, and finally by ultrathin sections without highly developed instruments and sophisticated techniques. Both methods proved that only the colocalization of a positive immunoreaction on the structure of interest (i.e., viruses) and identification of this structure on an ultrastructural level gives indisputable proof of the presence of virions.

## Figures and Tables

**Figure 1 viruses-13-01816-f001:**
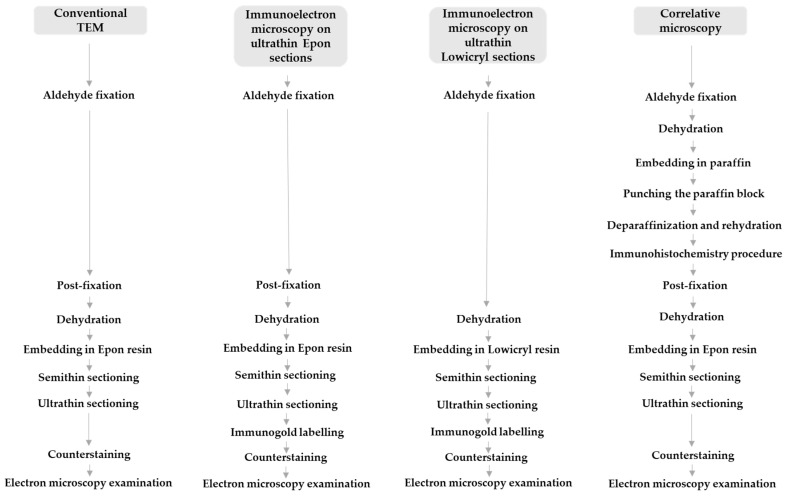
Sample preparation procedures for conventional transmission electron microscopy (TEM), correlative microscopy, and immunoelectron microscopy.

**Figure 2 viruses-13-01816-f002:**
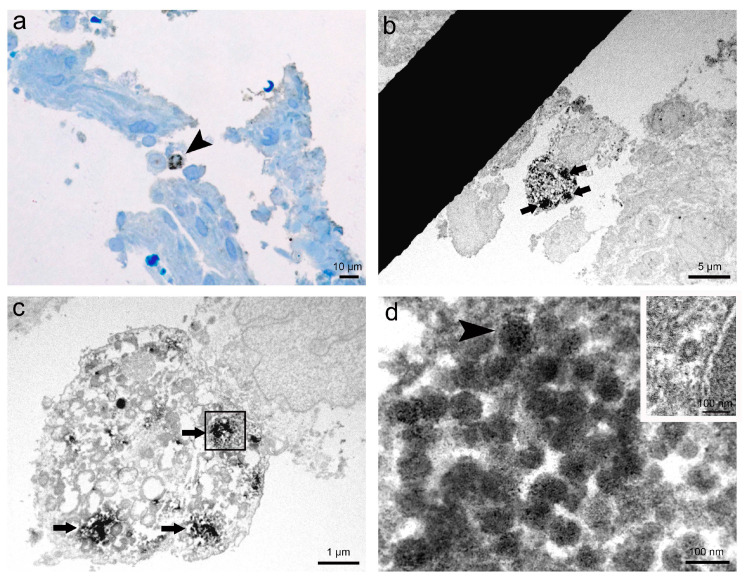
Representative images of immunohistochemistry and correlative microscopy of SARS-CoV-2 virions in nasopharyngeal tissue samples. (**a**) Micrograph of a semithin section of nasopharyngeal tissue immunolabelled against SARS-CoV-2 nucleocapsid protein. Note the cell (arrowhead) with clearly visible positive IHC reaction, shown as brown stained spots in the cytoplasm. Azure II staining; (**b**) TEM micrograph of the selected area of the semithin section with the same immunolabelled cell from image a. Note the electron-dense reaction product of DAB in the cytoplasm (arrows); (**c**) Higher magnification TEM micrograph of the cell (from image b) with clusters of the electron-dense reaction product of DAB in the cytoplasm (arrows) due to positive IHC reaction; (**d**) High magnification TEM micrograph of endosome (in a boxed area on image c) containing SARS-CoV-2 virions marked with the fine granular electron-dense reaction product of DAB. Note viral spike (S) glycoproteins on the surface of the virion (arrowhead). In the upper right inset is an unlabelled clathrin-coated vesicle, proving IHC reaction specificity to SARS-CoV-2 virions.

**Figure 3 viruses-13-01816-f003:**
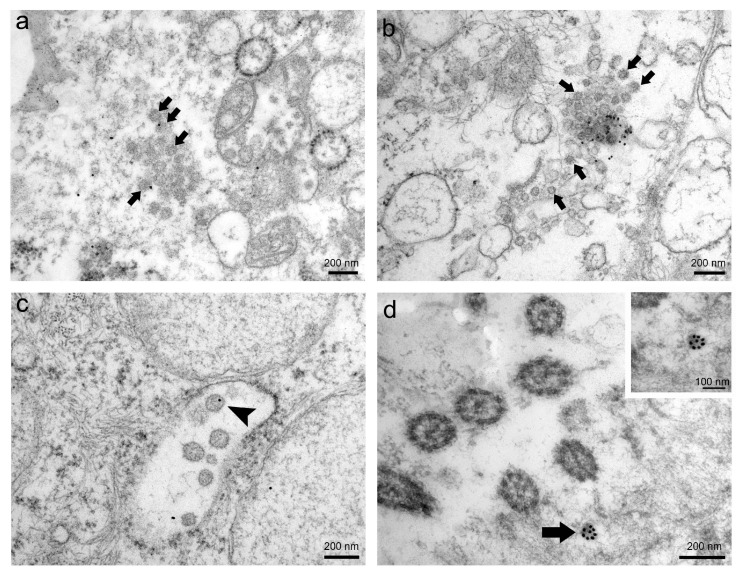
Representative images of immunogold labelling of SARS-CoV-2 virions in nasopharyngeal tissue samples. (**a**) TEM micrograph of ubiquitously scattered and unspecific immunogold labelling (gold particles) on ultrathin Epon section against SARS-CoV-2 nucleocapsid protein. Arrows indicate presumed virions. (**b**) TEM micrograph of scarce and unspecific immunogold labelling (gold particles) on ultrathin Epon section against SARS-CoV-2 spike glycoprotein S1. Arrows indicate presumed virions. (**c**) TEM micrograph of scarce but specific immunogold labelling against SARS-CoV-2 spike glycoprotein S1 after antigen unmasking procedure on ultrathin Epon section. Note that only individual virions are gold-labelled (arrowhead). (**d**) TEM micrograph of intense and specific immunogold labelling against SARS-CoV-2 spike glycoprotein S1 on ultrathin Lowicryl section. Note SARS-CoV-2 virion heavily marked with gold particles (arrow). The same virion is magnified in the upper right inset.

## Data Availability

Not applicable.

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
