# Peer review of "Just Seeing Is Not Enough for Believing: Immunolabelling as Indisputable Proof of SARS-CoV-2 Virions in Infected Tissue"

_viruses, 2021, doi:10.3390/v13091816_

Round 1

Reviewer 1 Report

Overall quite well-written.

Remember to define the non-standard abbreviations before first use, e.g. IHK, especially in the abstract. 

Please consider citing 

  1. Clin Microbiol Rev . 2009 Oct;22(4):552-63.
  2. J Eur Acad Dermatol Venereol. 2021 May 25:10.

and discuss the significance of the current method in relation to existing methods used for confirmation of viral detection in histological specimen. 

Author Response

Reviewer 1

Comment 1: Remember to define the non-standard abbreviations before first use, e.g. IHK, especially in the abstract. 

Author response:

We kindly thank the Reviewer for this observation. Following the Reviewer's suggestion, we have defined abbreviations where needed. Please find the corrections highlighted in yellow in the revised version of our manuscript.

Comment 2: Please consider citing 

  1. Clin Microbiol Rev . 2009 Oct;22(4):552-63.
  2. J Eur Acad Dermatol Venereol. 2021 May 25:10.

and discuss the significance of the current method in relation to existing methods used for confirmation of viral detection in histological specimen. 

Author response:

We took into consideration the Reviewer’s suggestion and we have included suggested articles and discussed the significance of our method in comparison to existing methods in the manuscript. Please find these changes highlighted in yellow in the revised version of the manuscript (page 2).

Reviewer 2 Report

This manuscript focuses on establishing a combination of immunohistochemistry with electronic microscopy for identifying the presence of SARS-CoV-2 virions in patients' tissues. It is a short and straightforward article or more like a protocol for IEM. I appreciate the details of methodology given by the authors but the scientific output is relatively limited. Perhaps this manuscript is better fit in journals related to methodology or protocol. This notwithstanding, hereunder is my comments concerning the content: 

Comments: 

1.) SARS nucleocapsid protein. (SARS is not the name of the virus, please revise all as SARSr-CoV)

2.)  Better to include the scale bars within each figure instead of figure legends. 

3.) It is confusing for your first discussion statement. You mention viral genetic material analysis is NOT enough for proving the SARS-CoV-2 in tissue due to the TEM misleading feature. I don't see any logical correlation within this sentence. Not to mention that IHC or even IEM also has same limitation as TEM like misleading interpretation or Ab cross reactivity. Then in authors' sense, what should be the golden standard in identifying the presence of viruses in particular tissue and does it synchronize with the global medical laboratory standard? 

4.) The authors should make a clear list, summary or table to distinguish out what are the differences between the proposed methodology with the traditional TEM method. 

Author Response

Reviewer 2

Comment 1: SARS nucleocapsid protein. (SARS is not the name of the virus, please revise all as SARS-CoV)

Author response:

We thank the Reviewer for this observation. We have corrected the name of the SARS-CoV-2 virus throughout the text. Please find our corrections highlighted in yellow in the revised version of the manuscript.

Comment 2: Better to include the scale bars within each figure instead of figure legends. 

Author response:

Following the Reviewer's suggestion, we have included the length of the scale bar within each figure. Please find the corrected figures in the revised version of our manuscript.

Comment 3: a) It is confusing for your first discussion statement. You mention viral genetic material analysis is NOT enough for proving the SARS-CoV-2 in tissue due to the TEM misleading feature. I don't see any logical correlation within this sentence.

Author response:

We appreciate the Reviewer's observation of this confusing first sentence of the Discussion. For better clarity, we have corrected the beginning of the Discussion and it is now as follows:

Viral genetic material analysis of patient’s nasopharyngeal tissue by itself is not sufficient proof of SARS-CoV-2 active infection in body tissues, but positive RT-PCR test, which is currently established test for SARS-CoV-2 infection is a precondition for further analysis of SARS-CoV-2 virus presence in a particular tissue. Likewise, TEM analysis per se is not enough for unequivocal proof of SARS-CoV-2 virions in the tissue of interest due to the similar structure and size of these virions and ubiquitous cell structures [4,5]. Thus we emphasized in a previous study that after positive RT-PCR test for SARS-CoV-2 RNA, IHC or even immunoelectron microscopy are needed for the reliable identification of virions in suspected infected tissues [6].

Please find the corrected first paragraph of the Discussion section highlighted in yellow in the revised manuscript.

  1. b) Not to mention that IHC or even IEM also has same limitation as TEM like misleading interpretation or Ab cross reactivity. Then in authors' sense, what should be the golden standard in identifying the presence of viruses in particular tissue and does it synchronize with the global medical laboratory standard? 

Author response:

We believe that immunoelectron microscopy is the most reliable method for distinguishing intracellular viral particles from normal cell structures of similar morphology and size as virions. Unfortunately, special and expensive equipment is needed for this methodology. Therefore, we developed a variant of correlative microscopy that allows every routine pathology laboratory to check its own results of IHC performed on routinely used paraffin-embedded samples also with the transmission electron microscope without highly developed instruments and sophisticated techniques. Namely, IHC reaction on light microscopy level per se could be misleading due to various factors in pre-analytical phase and misinterpretation of nonspecific staining (IHC must be thoroughly evaluated and validated by using negative controls and by testing antibody cross-reactivities to avoid false positive results). Additionally, low magnification and resolution of IHC does not allow pathologists to see virions and be sure that IHC reaction colocalizes with virions. Likewise, conventional TEM analysis per se is also not enough (as we stated in the title of our manuscript) for undoubted proof of present virions. Namely, despite the possibility to see the structures (virions) with TEM due to its high resolution, we can misinterpret them. So only the combination of the localization of a positive immunoreaction on the structure of interest (i.e., viruses) and simultaneous identification of this structure on an ultrastructural level gives indisputable proof of the presence of virions. We could therefore recommend pathologists in routine pathology laboratories that their results of IHC against the SARS-CoV-2 virus confirm with correlative microscopy described in our manuscript. To our opinion, this is the main scientific output of our work that can be transferred into everyday practice. We are aware that it represents a small piece of SARS-CoV-2 jigsaw but every piece is important in making a whole picture clearer. We cannot afford false positive or false negative pathology reports these days, especially if we have appropriate knowledge and methodology.

Comment 4: The authors should make a clear list, summary or table to distinguish out what are the differences between the proposed methodology with the traditional TEM method. 

Author response:

As was suggested by the Reviewer, we have made a table with the main steps of sample preparation procedures for conventional transmission electron microscopy (TEM), correlative microscopy, and immunoelectron microscopy, where the differences between all four methodologies are nicely seen. Please find this Table 1 in the revised version of the manuscript.

Round 2

Reviewer 2 Report

Reply to 3b is quite inspiring. I suggest the authors may consider to include this in their discussion part for better clarification. 

Table 1 is informative but not reader friendly (distracting). Authors may consider to redesign the table for better illustration of what message they would like to deliver. 

Author Response

Reviewer 2

Comment 1: Reply to 3b is quite inspiring. I suggest the authors may consider to include this in their discussion part for better clarification. 

Author response:

Following the Reviewer's suggestion, we have included the majority of the answer to the Disussion. Please find our corrections highlighted in yellow in the revised version of our manuscript.

From our point of view, immunoelectron microscopy is the most reliable method for distinguishing intracellular viral particles from normal cell structures of similar morphology and size as virions. Unfortunately, special and expensive equipment is needed for this methodology. Therefore, we developed a variant of correlative microscopy that allows every routine pathology laboratory to check its results of IHC performed on routinely used paraffin-embedded samples also with the transmission electron microscope without highly developed instruments and sophisticated techniques. Namely, IHC reaction on light microscopy level per se could be misleading due to various factors in the pre-analytical phase and misinterpretation of nonspecific staining. Additionally, low magnification and resolution of IHC do not allow pathologists to see virions and be convinced that IHC reaction colocalizes with virions. Likewise, conventional TEM analysis per se is also not enough for undoubted proof of present virions. Namely, despite the possibility to see the structures (virions) with TEM due to its high resolution, we can misinterpret them. So only the combination of the localization of a positive immunoreaction on the structure of interest (i.e., viruses) and simultaneous identification of this structure on an ultrastructural level gives indisputable proof of the presence of virions. We could therefore recommend pathologists in routine pathology laboratories to confirm their results of IHC against the SARS-CoV-2 virus with correlative microscopy described in our manuscript. To our opinion, this is the main scientific output of our work that can be transferred into everyday practise. We are aware that it represents a small piece of SARS-CoV-2 jigsaw but every piece is important in making a whole picture clearer. We cannot afford false positive or false negative pathology reports these days, especially if we have appropriate knowledge and methodology.

Comment 2: Table 1 is informative but not reader friendly (distracting). Authors may consider to redesign the table for better illustration of what message they would like to deliver. 

Author response:

We thank the Reviewer for this observation. We have corrected the Table 1 as suggested. Please find our corrections highlighted in yellow in the revised version of the manuscript.

Conventional TEM

Immunoelectron microscopy on ultrathin Epon  sections

Immunoelectron microscopy on ultrathin Lowicryl sections

Correlative microscopy

Aldehyde fixation

Aldehyde fixation

Aldehyde fixation

Aldehyde fixation

Dehydration

Embedding in paraffin   

Punching the paraffin block

Deparaffinization and rehydration

Immunohistochemistry procedure

Post-fixation

Post-fixation

Post-fixation

Dehydration

Dehydration

Dehydration

Dehydration

Embedding in Epon resin

Embedding in Epon resin

Embedding in Lowicryl resin

Embedding in Epon resin

Semithin sectioning

Semithin sectioning

Semithin sectioning

Semithin sectioning

Ultrathin sectioning

Ultrathin sectioning

Ultrathin sectioning

Ultrathin sectioning

Immunogold labelling

Immunogold labelling

Counterstaining

Counterstaining

Counterstaining

Counterstaining

Electron microscopy examination

Electron microscopy examination

Electron microscopy examination

Electron microscopy examination
